# Facile and Safe Synthesis of Novel Self-Pored Amine-Functionalized Polystyrene with Nanoscale Bicontinuous Morphology

**DOI:** 10.3390/ijms21249404

**Published:** 2020-12-10

**Authors:** Qilin Gui, Qi Ouyang, Chunrong Xu, Hongxue Ding, Shuxian Shi, Xiaonong Chen

**Affiliations:** 1Beijing Laboratory of Biomaterials, Beijing University of Chemical Technology, Beijing 100029, China; guiql@mail.buct.edu.cn (Q.G.); ouyangqi0000@163.com (Q.O.); Dhx1638170228@163.com (H.D.); shisx@mail.buct.edu.cn (S.S.); 2Key Laboratory of Carbon Fiber and Functional Polymers, Ministry of Education, Beijing University of Chemical Technology, Beijing 100029, China; badwpww@163.com

**Keywords:** *N*-vinylformamide, free-radical polymerization, self-pored, functionalized polystyrene, bicontinuous morphology

## Abstract

The chloromethyl-functionalized polystyrene is the most commonly used ammonium cation precursor for making anion exchange resins (AER) and membranes (AEM). However, the chloromethylation of polystyrene or styrene involves highly toxic and carcinogenic raw materials (e.g., chloromethyl ether) and the resultant ammonium cation structural motif is not stable enough in alkaline media. Herein, we present a novel self-pored amine-functionalized polystyrene, which may provide a safe, convenient, and green process to make polystyrene-based AER and AEM. It is realized by hydrolysis of the copolymer obtained via random copolymerization of *N*-vinylformamide (NVF) with styrene (St). The composition and structure of the NVF-St copolymer could be controlled by monomeric ratio, and the copolymers with high NVF content could form bicontinuous morphology at sub-100 nm levels. Such bicontinuous morphology allows the copolymers to be swollen in water and self-pored by freeze-drying, yielding a large specific surface area. Thus, the copolymer exhibits high adsorption capacity (226 mg/g for bisphenol A). Further, the amine-functionalized polystyrene has all-carbon backbone and hydrophilic/hydrophobic microphase separation morphology. It can be quaternized to produce ammonium cations and would be an excellent precursor for making AEM and AER with good alkaline stability and smooth ion transport channels. Therefore, the present strategy may open a new pathway to develop porous alkaline stable AER and AEM without using metal catalysts, organic pore-forming agents, and carcinogenic raw materials.

## 1. Introduction

Anion exchange resins (AER) and anion exchange membranes (AEM) have always attracted much attention and are widely used in many fields, such as water treatment [1,2], medicine purification [3], microbial or anion exchange membrane fuel cell [4,5,6,7,8], and catalysis. For many decades, benzyltrimethylammonium, which is easily synthesized by quaternization of the chloromethyl-functionalized polystyrene, has been the most commonly used cation in polystyrene-based AEM and AER (Scheme 1I) [7,9]. However, this structural motif is not stable enough in alkaline media and can be degraded through β-elimination reaction at the benzyl position [7,10]. In addition, the chloromethylation of polystyrene or styrene involves highly toxic and carcinogenic raw materials (e.g., chloromethyl ether and dichloromethyl ether), which are not environmentally friendly, and has caused many safety concerns [11,12,13]. In addition, AER and AEM made by this method remain with a number of inherent shortcomings, including poor adjustability of amino-moiety, self-crosslinking side reaction, potential metal catalyst pollution, and usage of toxic and highly inflammable porogens such as toluene, xylene, dichloroethane, and aliphatic hydrocarbons (Scheme 1I) [12,13,14]. Therefore, it is necessary to develop a safe and green procedure to make AER and AEM. 

Recently, it has been generally accepted that the all-carbon backbone and alkyltrimethylammonium cations are still the most promising structural motifs for alkaline stable AEM [7,15]. Amine-functionalized polystyrene (PSt-NH_2_) may provide an alternative way for manufacturing AEM and AER, since the amine groups on polymer side chains can be easily quaternized. Thus, the PSt-NH_2_ prepared by copolymerization of styrene (or styrene derivatives) and *N*-vinylformamide (NVF) may open up a new strategy for green manufacturing of alkaline stable polystyrene-based AER and AEM. On the one hand, NVF units in the copolymer could be easily hydrolyzed to produce amine groups, which are on the polymer side chain and easily converted into alkyltrimethylammonium cations by quaternization [16,17]. The manufacturing process is environmentally friendly, since it involves no metal catalysts, toxic porogens, or carcinogenic raw materials. On the other hand, AEM made by the PS-NH_2_ may have good alkaline stability because of its all-carbon backbone structure and alkyltrimethylammonium cations. Although dimethylaminoethyl methacrylate could also provide amine groups, the ester structure is unstable under acidic or basic conditions and the amine group would be easily lose by the hydrolysis of the ester. Further, unlike its isomer acrylamide, NVF is a non-toxic monomer and causes less safety concern [18,19]. 

Unfortunately, successful random copolymerization of NVF with styrene has rarely been reported, although NVF has high polymerizing reactivity [20], and can randomly copolymerize with vinyl monomers such as acrylic acid, *N*-vinylpyrrolidone, and acrylamide [21,22]. On the one hand, the conjugation effect of styrene is obviously different from that of NVF. The Q values of NVF and styrene are 1.0 and 0.29, respectively [23]. Such a large difference in Q value increases the difficulty of copolymerization. On the other hand, the polarities of NVF and styrene are so different that only a few solvents could dissolve both monomers and their polymers. Polystyrene microspheres with poly(*N*-vinylformamide) (PNVF) shell were prepared by emulsion polymerization or suspension polymerization [24,25,26]. However, there was no evidence to show random links between the NVF unit and the styrene unit, and the NVF content in those polymer particles was low (<5%).

In this article, the free-radical copolymerization of styrene and NVF was performed using AIBN as the initiator and DMF as the solvent. We demonstrated for the first time that NVF could randomly copolymerize with styrene. Copolymers with adjustable monomeric composition were successfully obtained via adjusting the monomer feed ratio and monomer concentration. Interestingly, a bicontinuous morphology at sub-100 nm level was found in copolymers with high NVF mass percentage. Such bicontinuous morphology allowed NVF-St copolymers to be well swollen in water to facilitate the exposure and hydrolysis of NVF units (Figure 1a). Thus, the porous PSt-NH_2_ with different amine content was obtained by hydrolysis of NVF-St copolymers. Such PSt-NH_2_ could be further quaternized to make polystyrene-based AER and AEM with all-carbon backbone structure and alkyltrimethylammonium cations (Scheme 1II). Additionally, the preparation process did not involve any metal catalyst, toxic porogen, or carcinogenic raw material. Therefore, the preparation of PSt-NH_2_ by hydrolyzing the copolymer of styrenic monomers and NVF provided a new strategy for the green production of alkaline and stable polystyrene-based AER and AEM.

## 2. Results and Discussion

### 2.1. Structure and Composition of the NVF-St Copolymer

Fourier transform infrared spectroscopy (FTIR) and nuclear magnetic resonance spectroscopy (NMR) were conducted to reveal the structure and composition of the copolymerizing products. As shown in Figure 1a, the spectrum of copolymerizing product (P(NVF-St)) showed a strong C = O stretching at 1670 cm^−1^ and a phenyl ring C-H bending signal at 760 cm^−1^, which were contributed by NVF and styrene units, respectively [27,28,29]. In the ^1^H-NMR spectra of P(NVF-St) samples (Figure 1b), peak **a** at 7.5–8.1 and peak **b** at 6.4–7.3 ppm could be assigned to the protons (1H) of aldehyde group of NVF units and the aromatic protons (5H) of styrene units, respectively [19,30,31]. Thus, both IR and NMR spectra confirmed the formation of NVF-St copolymers.

Moreover, the intensity of peak **a** showed a positive correlation of NVF content in the copolymer (***F*_NVF_**) to the NVF monomer content in the polymerization feed (***f*_NVF_**). Similarly, an increase in the total monomer concentration led to an increase in ***F*_NVF_** (Appendix A). Therefore, copolymers with different NVF content could be obtained by adjusting the ***f*_NVF_** and total monomer concentration. The ***F*_NVF_** was quantitatively calculated according to the intensity of the characteristic peaks **a** and **b** in the NMR spectra. As shown in Table 1, although the ***F*_NVF_** was always smaller than ***f*_NVF_** for all the recipes investigated, the ***F*_NVF_** increased alongside the increase of the ***f*_NVF_**, suggesting a statistical model or random style of the copolymerization (Appendix A). The monomer reactivity, r_1_ (NVF) and r_2_ (St) were roughly estimated to be 0.34 and 10.1, respectively, based on the fitting plot of ***F*_NVF_** against ***f*_NVF_** (Appendix A). According to the Q and e values of the two monomers, the theoretically predicted reactivity r_1_ and r_2_ were 0.34 and 2.76, respectively [23]. The real monomer reactivity showed some difference from that calculated using Q and e values.

Furthermore, the glass transition temperature (*T*_g_) of the obtained copolymers was measured by differential scanning calorimetry (DSC), which could be used to distinguish random copolymers from polymer blends and block copolymers. Generally, only a single *T*_g_ could be detected for the homo-polymer or random copolymer, while two or more *T*_g_ values could be observed for blended polymers or block copolymers [32]. As shown in Figure 2a, the homo-polystyrene (PSt) exhibited a glass transition around 95 °C, while the PNVF was around 142 °C. There were two *T*_g_ values at about 95 and 142 °C in the DSC curve of the PNVF/PSt blend (Figure 2a), corresponding to PSt and PNVF, respectively. Notably, Figure 2b and Appendix A show that only one *T*_g_ appeared in each DSC curve of the copolymer samples. These results obtained from glass transition measurement confirmed random architecture of the NVF-St copolymers. Table 1 provides all *T*_g_ values of the copolymers, showing that the *T*_g_ of the copolymer increased with the increase of the ***F*_NVF_**. Thus, the Fox equation [32,33], which is an empirical formula to describe the linear dependence of the glass transition temperature of a random copolymer on its composition, was used to reveal the relationship between the *T*_g_ and the composition of the NVF-St copolymers.
(1)$$\frac{1}{T_{g}}=\left(\frac{1}{T_{gA}}-\frac{1}{T_{gB}}\right)\omega_{A}+\frac{1}{T_{gB}}$$
where, *T*_gA_, *T*_gB_, and *T*_g_ are the glass transition temperatures of PNVF (142 °C), PSt (95 °C), and NVF-St copolymer, respectively, and $\omega_{A}$ is the mass percentage of NVF unit in the copolymers (***F*_NVF_**). Figure 2c presents the plot of *T*_g_^−1^ against the $\omega_{A}$(***F*_NVF_**). The experimental plot showed good linearity (with a correlation coefficient of 0.9906) and was close to the plot using the Fox equation. This result well demonstrated random architecture of the NVF and St units composing the copolymer. Notably, the *T*_g_ of NVF-St copolymer was larger than that of PSt, suggesting a better heat resistance of the functionalized polystyrene than the homo-polystyrene. In the future, we may prepare copolymers with higher glass transition temperature and good heat resistance through copolymerization of styrene derivatives and NVF.

### 2.2. Microphase Separation Morphology of the NVF-St Copolymer

Styrene is known as the most commonly used hydrophobic monomer, while NVF is a hydrophilic functional monomer [21]. Thus, a hydrophobic phase (dominated by styrene units) and a hydrophilic phase (dominated by NVF units) could be formed in an interpenetrating morphology [6]. Similarly, phase-separated co-network morphology was observed in the case of copolymer of poly(dimethylsiloxane)-α,ω-diacrylate and N,N-dimethylacrylamide [34]. It seems that the significant difference in polarity of the two monomeric moieties is responsible for the immiscibility at the microscale. To visually study the morphology of the sample, atomic force microscopy (AFM) was used. AFM images confirmed that the NVF-St copolymer underwent phase separation and formed a two-phase bicontinuous structure with non-long-range ordering, when the ***F*_NVF_** became larger than 50% (Figure 3). Note that the size of the bicontinuous morphology was sub-100 nm, which was much smaller than the classical one formed by block copolymers or blends [35,36]. This may have been caused by the random distribution of NVF and styrene units, which prevented the growth of both hydrophobic and hydrophilic phases by forming an interpenetrated phase separation morphology.

To further verify the microphase separation structure of the copolymer, transmission electron microscopy (TEM) was used. The copolymer samples were annealed at 120 °C for 3 h, and dyed by reaction of –NH-CHO groups in NVF units with RuO_4_. Figure 4 shows that as the ***F*_NVF_** increased, the area of the dark black zone became larger, indicating that the hydrophilic phase dominated by NVF increased. Moreover, when the ***F*_NVF_** increased from 37.4% to 58.0%, the copolymer underwent a phase transition and formed a microphase separation morphology. The NVF-St-58.0 exhibited obvious an microphase separation phenomenon between a hydrophilic phase dominated by NVF units (dark black) and a hydrophobic phase dominated by styrene units (light gray) due to the self-assembly behavior from two different kinds of incompatible segments in the copolymer. In the TEM images, the characteristic size of the microphase separation structure was about 40 nm, which was consistent with the results of AFM. It is worth mentioning that there were dense black spots at the hydrophilic phase and the hydrophilic/hydrophobic phase interface, indicating that the NVF side groups (-NH-CHO) in the copolymer could be exposed in water. Thus, the bicontinuous morphology may have facilitated the exposure and hydrolysis of NVF units to generate amine groups, which was essential for the preparation of amine-functionalized polystyrene. In addition, the good film-forming performance of the NVF-St copolymer allowed it to be possible to prepare the copolymer membrane, which could be converted into PSt-NH_2_ membrane by hydrolysis and further converted into AEM through quaternization.

### 2.3. Hydrolysis, Quaternization, and Potential Applications of the NVF-St Copolymer

The hydrolysis and further quaternization of the NVF-St copolymer were carried out. Comparing the IR spectra of P(NVF-St) and PSt-NH_2_, it was found that a strong amine absorption peak appeared at 1452 cm^−1^, while the aldehyde absorption peak at 1670 cm^−1^ almost disappeared (Figure 5a). This indicated that NVF-St copolymers could easily hydrolyze under basic conditions to produce PSt-NH_2_ (Scheme 1II). The resultant PSt-NH_2_ could be quaternized using bromoethane under the mild condition, as shown by the peak at 1114 cm^−1^, corresponding to the C–N bond of quaternary ammonium in the IR spectra (Figure 5a) [28,37]. It is known that a quaternary ammonium containing benzyl is not stable under strong alkali conditions due to β-elimination reaction [7]. Compared with ammonium cations prepared from chloromethyl-functionalized polystyrene, the alkyltrimethylammonium cations prepared based on the PSt-NH_2_ may have had better alkaline stability because the resultant quaternary ammonium moieties did not contain benzyl group. Moreover, the all-carbon skeleton structure could also improve the alkaline stability of AEM and AER [15]. Therefore, the PSt-NH_2_ could be used as the precursor for making alkaline stable polystyrene-based AER and AEM.

Functionalized polystyrene is a commonly used adsorbent; thus, it is worth evaluating the adsorption performance of the NVF-St copolymers before and after hydrolysis. Bisphenol A (BPA), which has been intensively concerned as an endocrine disruptor [38], was selected as a model pollutant. Figure 5b,c shows that the BPA uptake (mg/g) of the NVF-St copolymer significantly increased when the ***F*_NVF_** reached 50%. This is because the NVF-St copolymer underwent a phase separation and formed a bicontinuous morphology at the specific ***F*_NVF_**. The bicontinuous morphology allowed the NVF-St copolymer to be swollen in water and self-pored by freeze-drying. Therefore, the freeze-dried copolymer provided improved adsorption performance (Figure 5b). The mechanism of self-poring by the freeze-drying process is shown in Scheme 1. The hydrophilic phase dominated by NVF units in the copolymer sample absorbed and swelled in water. The absorbed water was frozen and removed during freeze drying, leading to the formation of pores. In such a case, water serves as a pore-forming agent. Compared with organic porogens, water is a safe and cheap porogen, which may provide a green and safe pore-forming process. 

Scanning electron microscopy (SEM) and Brunaner–Emmett–Teller (BET) analysis were used to observe the formation of holes. SEM images (Figure 6) and the significant increase in specific surface area (Appendix A) also demonstrated the formation of through-holes, which was related to the hydrophilic–hydrophobic bicontinuous morphology. According to Figure 6a,b, the sample (No.6) prepared under vacuum-dried had smooth and continuous surface morphology, while after freeze-drying, the morphology became porous and rough. Therefore, the specific surface area (Appendix A) of the latter (80.5 m^2^/g) was much larger than that of the former (3.5 m^2^/g). The specific surface area directly affected the adsorption performance. In future study, the pore morphology and specific surface area of the NVF-St copolymer may be further well controlled by adjusting the conditions such as the freezing rate. Most importantly, the bicontinuous morphology facilitated the exposure and hydrolysis of NVF units to generate amine groups (Figure 5a). After hydrolysis, the sample (No.6) showed the highest BPA uptake (as high as 86 mg/g and about 8.5 times larger than that of PSt, Figure 5b). The hydrolyzed product not only had a porous structure (Figure 6c), but also provided an additional positive charge to interact with the phenol group in BPA. Additionally, the increase of the BPA equilibrium concentration could further improve the BPA uptake of the sample. When the equilibrium concentration was increased to 0.5 mmol/L, the BPA uptake of the hydrolyzed product reaches 226 mg/g. Compared with commonly used activated carbon and polymer adsorbents (Table 2), NVF-St copolymer had higher adsorption capacity and could be prepared in a facile way by free radical copolymerization. Therefore, the strategy of free-radical copolymerization of NVF with styrene may be used to obtain PSt-NH_2_ in a simple way. In short, the resultant PSt-NH_2_ was a desirable precursor for making AER and AEM, because it had adjustable water uptake, obvious microphase separation structure, and green and facile preparation process.

## 3. Materials and Methods 

### 3.1. Materials and Instrumentations

Materials and instrumentations are shown in the Appendix A. *N*-vinylformamide (NVF) (Aldrich) and styrene (St) (Beijing Chemical Works, Beijing, China) were purified by distillation under reduced pressure. The structure and composition of the NVF-St copolymer were characterized by Fourier transform infrared spectroscopy (FTIR), nuclear magnetic resonance spectroscopy (NMR), differential scanning calorimetry (DSC), and gel permeation chromatography (GPC). The morphology and self-poring mechanism were revealed by atomic force microscope (AFM), transmission electron microscope (TEM), scanning electron microscopy (SEM), and Brunaner–Emmett–Teller (BET) analysis. The DMSO solution of NVF-St copolymer (1–3 mg/mL) was dropped on a silicon wafer to form a film by KW4A spin coater at 3000 r/min for 60 s. Samples were annealed at 120 °C for three hours before testing AFM. UV-visible spectroscopy was used to determine the concentration of BPA in solution.

### 3.2. Experimental Procedure

#### 3.2.1. Copolymerization of NVF and St

The copolymerization was performed under nitrogen atmosphere at 60 °C in a 50-mL sealed flask for 10 h using AIBN (1.5% of the total mass of the monomers) as the initiator and DMF as the solvent. Different mass ratios of monomers ((NVF)/(St) were 70/30, 83/17, 88/12, 90/10, respectively) were employed and the total monomer concentration controlled as 20%, 30%, or 40%. The copolymerizing product was dissolved in a mixed solvent of DMSO and water in a volume ratio of DMSO/water at 3/1. Then ethanol was added into the solution to obtain polymer precipitation. Such dissolution/precipitation was repeated three times. The resultant final precipitate was vacuum-dried or freeze-dried until a constant weight was reached, to obtain pure copolymer (yield 30–50%). The molecular weight of the copolymer (No.1) was found to be around 12 kDalton (Appendix A).

#### 3.2.2. Hydrolysis of the NVF-St Copolymer

The copolymer was hydrolyzed to obtain polyvinylamine. Typically, in a flask, 4.0 g of the copolymer (No.6) was mixed with 15 mL of water and 35 mL of 5% aqueous NaOH solution. The mixture was magnetically stirred and heated in a 60 °C oil bath for 24 h. The resultant solution was neutralized to pH 7.0 using 1 mol/L HCl aqueous solution, followed by dialyzing and freeze-dried to obtain 2.9 g product (the degree of hydrolysis: 85%).

#### 3.2.3. Quaternization of the Hydrolyzed Copolymer

Typically, in a flask, 1.4 g hydrolyzed copolymer (No.6), 8.6 g bromoethane, and 6.3 g Na_2_CO_3_ were mixed with 30 mL water and 30 mL DMSO. The mixture was magnetically stirred and heated in a 60 °C oil bath for 8 h. The resultant solution was neutralized using 1 mol/L HCl aqueous solution. Then ethanol was added into the mixture to obtain polymer precipitation. The polymer precipitation was dialyzed and freeze-dried to obtain 1.2 g product.

#### 3.2.4. Batch Adsorption Experiments

Adsorption studies were performed in 10-mL centrifuge tubes equipped with a magnetic stir bar (the stirring rate is 150 r.p.m.) at ambient temperature. The NVF-St copolymer (10 mg) was added into 5 mL of BPA solution (200 mg/L, pH 7) for 18 h. Then, the mixture was filtered by a PTFE (0.2 μm) membrane, and the residual concentration of BPA in each sample was determined by UV-visible spectroscopy. The amount of BPA bound to the copolymer ($Q_{m}$) was calculated by the following equation:(2)$$Q_{m}=\frac{\left(C_{o}-C_{e}\right)\times V}{m_{c}}$$
where $C_{o}$ and $C_{e}$ are the BPA concentrations in solution before and after adsorption, respectively, *V* is the volume of the solution, and $m_{c}$ is the mass of the copolymer.

## 4. Conclusions

In summary, NVF-St copolymers with different NVF contents were synthesized via free-radical copolymerization in a simple way by adjusting the monomer feed ratio and monomer concentration. The bicontinuous morphology with a size at sub-100 nm level was formed in the NVF-St copolymers with high NVF mass percentage. Such a structure allowed NVF-St copolymers to be self-pored by water-swelling and freeze-drying, resulting in a large specific surface area. The BPA uptake of the copolymer reached 226 mg/g, which was larger than that of the commonly used activated carbon and polymer adsorbents, indicating the copolymer has higher adsorption capacity. Additionally, the NVF-St copolymer was easily hydrolyzed to convert into amine-functionalized polystyrene, which could be further quaternized to produce ammonium cations. The amine-functionalized polystyrene would be an excellent precursor for making polystyrene-based AER and AEM with all-carbon backbone structure and benzyl-free quaternary ammonium and smooth ion transport channels. The entire process, from the copolymerization, hydrolysis, to the quaternization, did not involve any metal catalyst, toxic porogen, or carcinogenic raw material. Therefore, the process may provide a safe, green, and facile way to access AER and AEM. Prospectively, AER and AEM made by the amine-functionalized polystyrene may have potential applications in many fields, such as anion exchange membrane fuel cell, adsorbent materials, catalysts, water treatment, and drug separation/purification.

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
