# Peer review of "Facile and Safe Synthesis of Novel Self-Pored Amine-Functionalized Polystyrene with Nanoscale Bicontinuous Morphology"

_ijms, 2020, doi:10.3390/ijms21249404_

Round 1
Reviewer 1 Report
The submitted manuscript entitled "Facile and safe synthesis of novel self-pored amine-functionalized polystyrene with nanoscale bicontinuous morphology" deals with the synthesis of the novel an interesting material with application potential. In my opinion, the article in this form is publishable in the International Journal of Molecular Science after minor revision.
In Chapter 3. Materials and Methods please correct subcaption numbers.
Please consider adding the prepositions "of" to the experimental parts of the synthetic procedures (subchapters 3.2.3 and 3.2.4) as on lines 259-260. "Typically, in a flask, 4.0 g copolymer (No.6) was mixed with 15 mL water and 35 mL of 5% NaOH aqueous solution." replace with "Typically, in a flask, 4.0 g of the copolymer (No.6) was mixed with 15 mL of water and 35 mL of 5% aqueous NaOH solution.".
line 262 - "pH7.0" please add space pH 7.0
In my opinion, the authors should comment briefly on the use of carbon tetrachloride (line 254) in the experiment. The use of this substance somewhat undermines the idea of an environmentally friendly and safe approach.
Author Response
International Journal of Molecular Sciences (ijms-1022366)
TITLE: Facile and safe synthesis of novel self-pored amine-functionalized polystyrene with nanoscale bicontinuous morphology
Dear editor and reviewers:
Thanks for processing our manuscript. The comments and suggestions from reviewers are very valuable for revising the manuscript. We have completed a revision of the manuscript according to the suggestions. The following is our response or explain to the comments.
Let us know if the manuscript needs further revision.
Sincerely yours,
Xiaonong Chen
In Chapter 3. Materials and Methods please correct subcaption numbers.
Re: Thank you very much for the suggestion, we modify it in the revised text. (Page 9 of 13, Lines 235, 247, 248, 258, 264 and 270, red marked)
Please consider adding the prepositions "of" to the experimental parts of the synthetic procedures (subchapters 3.2.3 and 3.2.4) as on lines 259-260. "Typically, in a flask, 4.0 g copolymer (No.6) was mixed with 15 mL water and 35 mL of 5% NaOH aqueous solution." replace with "Typically, in a flask, 4.0 g of the copolymer (No.6) was mixed with 15 mL of water and 35 mL of 5% aqueous NaOH solution.".
Re: Thanks for the valuable comments. We modify it in the revised text. (Page 9 of 13, Lines 259-260, red marked)
line 262 - "pH7.0" please add space pH 7.0
Re: Thanks for the advice, we add it in the revised text. (Page 9 of 13, Line 262, red marked)
In my opinion, the authors should comment briefly on the use of carbon tetrachloride (line 254) in the experiment. The use of this substance somewhat undermines the idea of an environmentally friendly and safe approach.
Re: This is a good suggestion. We are also aware of this shortcoming. Recently we have modified the process. Ethanol is used to replace carbon tetrachloride. (Page 9 of 13, Line 254, red marked)

Reviewer 2 Report
The article by Chen and co-workers reports the free radical synthesis of copolymers of styrene with N-vinylformamide and preparation of porous structures from the resulting copolymers along with their application in adsorption of Bisphenol A.
I recommend publication after revision and addressing the following points:
-How does the authors make sure that all the monomers are consumed and converted to polymer? SEC data need to be provided.
-The fraction of each monomer needs to be calculated/ estimated for each polymer sample.
-I disagree with the authors in the statement that freeze-drying is a low cost and green process. This is an energy consuming method and also it is not applicable to large surface areas.
-The degree of hydrolysis is not stated. Authors need to calculated and verify the degree of hydrolysis.
-The authors need to state the point of using the vinylformamide. Why not use another common monomer bearing amine groups such as dimethyl ethyl methacrylate.
Author Response
International Journal of Molecular Sciences (ijms-1022366)
TITLE: Facile and safe synthesis of novel self-pored amine-functionalized polystyrene with nanoscale bicontinuous morphology
Dear editor and reviewers:
Thanks for processing our manuscript. The comments and suggestions from reviewers are very valuable for revising the manuscript. We have completed a revision of the manuscript according to the suggestions. The following is our response or explain to the comments.
Let us know if the manuscript needs further revision.
Sincerely yours,
Xiaonong Chen
How does the authors make sure that all the monomers are consumed and converted to polymer? SEC data need to be provided.
Re: Thanks for the valuable comments. SEC data of the copolymer (No.1) was provided in Fig. S4. Increasing the reaction time, and raising the temperature in the late stage of the polymerization could make more monomers be converted into polymers. Also, we are doing dispersion co-polymerization of NVF and St. By the dispersion procedure, the monomer conversion could be increased significantly within several hours. The co-polymerizing kinetics would be reported in a separated article.
-The fraction of each monomer needs to be calculated/estimated for each polymer sample.
Re: Thanks for the valuable comment. The fraction of NVF units in each polymer sample was calculated by NMR and shown in Table 1. The fraction of styrene units in each polymer sample can be obtained by the formula: FSt = 1- FNVF
-I disagree with the authors in the statement that freeze-drying is a low cost and green process. This is an energy consuming method and also it is not applicable to large surface areas.
Re: We understand the concerns raised by the reviewer in good faith. Environmental protection (reducing toxic pollutants) and energy consumption are complex topics. The aim our work is to provide an alternative approach that might become a potential consideration in making polystyrene-based ionic exchange materials. Compared with organic porogens, water is a safe and cheap porogen. The removal process of organic porogen also consumes energy. We hope our work would attract attention to fully evaluate the environmental impact of the presented approach in an industrial level..
-The degree of hydrolysis is not stated. Authors need to calculated and verify the degree of hydrolysis.
Re: This is a valuable and important comment. We add hydrolysis information to the revised manuscript (line 263). The degree of hydrolysis for sample (No.6) is about 85%.
-The authors need to state the point of using the vinylformamide. Why not use another common monomer bearing amine groups such as dimethyl ethyl methacrylate.
Re: Thanks for the valuable comment. On the one hand, NVF units in the copolymer could be easily hydrolyzed to produce amine groups. This is why NVF was developed and used as the precursor for making amide- and/or amine-functionalized polymers. On the other hand, for the dimethylaminoethyl methacrylate [we believe that the original intention of the reviewer is to recommend dimethylaminoethyl methacrylate, since dimethyl ethyl methacrylate does not contain amine group.], the amine moiety is connected to the resultant backbone upon copolymerization through an ester bond. The ester structure is unstable under acidic or basic condition and the amino group would be easily lose by the hydrolysis of the ester.
Reviewer 3 Report
In this work, Chen and coworkers report a new way of synthesizing anion exchange resins and membranes based on copolymerization of styrene and N-vinylformamide (NVF) monomers. Incorporation of NVF units, which could be further hydrolyzed to amine groups, provides a possibility for synthesizing AERs and AEMs upon quaternization under mild conditions. Due to the hydrophobic and hydrophilic nature of the monomers, the synthesized copolymers formed bi-continuous morphologies that were used as suitable materials for adsorbing pollutants such as BPA. The manuscript is written well and clear to understand. Further comments to consider:
- Copolymerization of St and NVF monomers was conducted in DMF, which is a commonly used solvent in polymerization reactions. However, the authors state that due to the differences in their polarity, no suitable solvent could dissolve these monomers and their polymers. This statement may be revised.
- Can the authors comment on the possibility of desorption of adsorbed BPA from the polymers and recycle them for further adsorption cycles?
- The term amine-functionalized polystyrene may be misleading, which might indicate functionalization of polystyrene through the phenyl rings. The authors may use a proper terminology that reflects the copolymer nature of PS with amine segments (PS-NH2).
- Results presented in Figure 5b and 5c show a maximum BPA uptake of ~80-85 mg/g for the FDH sample, which is significantly lower than the value of 226 mg/g discussed in the text. Can the authors clarify this discrepancy or explain in the text if the data are obtained under different conditions from different samples?
Minor comments:
- Line 62: ‘crylic acid’ should be ‘acrylic acid’
- Line 79: As quaternization is performed on primary amines using bromoethane, the term ‘alkyltrimethylammonium’ cations may be changed to quaternary ammonium cations
- The letter N in N-vinylformamide and poly(N-vinylformamide) should be Italic: N-vinylformamide and poly(N-vinylformamide)
- Line 111 and throughout the manuscript: glass transition temperature – Tg (only T italic)
- Line 112: change ‘form’ to ‘from’
Author Response
International Journal of Molecular Sciences (ijms-1022366)
TITLE: Facile and safe synthesis of novel self-pored amine-functionalized polystyrene with nanoscale bicontinuous morphology
Dear editor and reviewers:
Thanks for processing our manuscript. The comments and suggestions from reviewers are very valuable for revising the manuscript. We have completed a revision of the manuscript according to the suggestions. The following is our response or explain to the comments.
Let us know if the manuscript needs further revision.
Sincerely yours,
Xiaonong Chen
Copolymerization of St and NVF monomers was conducted in DMF, which is a commonly used solvent in polymerization reactions. However, the authors state that due to the differences in their polarity, no suitable solvent could dissolve these monomers and their polymers. This statement may be revised.
Re: Thanks for the valuable comments. We replace "almost no suitable solvent" with "only a few solvents ". (Page 2 of 13, Line 65-66, red marked)
Can the authors comment on the possibility of desorption of adsorbed BPA from the polymers and recycle them for further adsorption cycles?
Re: Thanks for the kind suggestion. The adsorbed BPA can be easily eluted by methanol, ethanol or basic aqueous solution based on our previous research on the membrane modified by grafting PNVF or its hydrolyzed chains (J. Mater. Chem. A. 8(32) (2020) 16487-16496 ). We are doing more work on the adsorption-desorption kinetics and the cycle performance of the St-NVF copolymer and the details will be reported in a separated manuscript.
Results presented in Figure 5b and 5c show a maximum BPA uptake of ~80-85 mg/g for the FDH sample, which is significantly lower than the value of 226 mg/g discussed in the text. Can the authors clarify this discrepancy or explain in the text if the data are obtained under different conditions from different samples?
Re: Thanks a lot for the advice. Yes, different condition leads to different results in the adsorption capability. The experimental condition for the adsorption capacity showed in Figure 5 (80-85mg/g) was: copolymer 10 mg, BPA 200 mg/L, 5 mL, pH 7, 18 h, room temperature. The experimental condition for the adsorption capacity 226mg/g was: copolymer 10 mg, BPA 200 mg/L, 30 mL, pH 7, 18 h, room temperature. More BPA added to the solution resulted in a greater adsorption. We clarify this in the revised text (Page 7 of 13, Line 220-222).
Minor comments:
Line 62: ‘crylic acid’ should be ‘acrylic acid’
The letter N in N-vinylformamide and poly(N-vinylformamide) should be Italic: N-vinylformamide and poly(N-vinylformamide)
Line 111 and throughout the manuscript: glass transition temperature – Tg (only T italic)
Line 112: change ‘form’ to ‘from’
Re: Thanks for the kind comments. We have made corrections in the revised text (red marked).

Round 2
Reviewer 2 Report
I think the authors have addressed all the raised points.
I suggest they incorporate their explanation/answer to questions 3 and 5 in the article text.
Publication after this minor correction is recommended.
Author Response
I suggest they incorporate their explanation/answer to questions 3 and 5 in the article text.
Re. Thanks for the valuable comments.
For questions 3, as shown in the manuscript (Page 8 of 14, Line 202-203, red marked), We have revised the description. The new content is: Compared with organic porogens, water is a safe and cheap porogen, which may provide a green and safe pore forming process.
For questions 5, We added a new description (Page 2 of 14, Line 58-60, red marked). The new content is:Although dimethylaminoethyl methacrylate could also provide amine groups, the ester structure is unstable under acidic or basic condition and the amine group would be easily lose by the hydrolysis of the ester.